# Novel Antimicrobial Peptides from Saline Environments Active against *E. faecalis* and *S. aureus*: Identification, Characterisation and Potential Usage

**DOI:** 10.3390/ijms241411787

**Published:** 2023-07-22

**Authors:** Jakub Lach, Magdalena Krupińska, Aleksandra Mikołajczyk, Dominik Strapagiel, Paweł Stączek, Agnieszka Matera-Witkiewicz

**Affiliations:** 1Department of Molecular Microbiology, Faculty of Biology and Environmental Protection, University of Lodz, 90-237 Lodz, Poland; jakub.lach@biol.uni.lodz.pl (J.L.); pawel.staczek@biol.uni.lodz.pl (P.S.); 2Biobank Lab, Department of Oncobiology and Epigenetics, Faculty of Biology and Environmental Protection, University of Lodz, 90-235 Lodz, Poland; dominik.strapagiel@biol.uni.lodz.pl; 3Screening of Biological Activity Assays and Collection of Biological Material Laboratory, Wroclaw Medical University Biobank, Faculty of Pharmacy, Wroclaw Medical University, 50-556 Wroclaw, Poland; magdalena.krupinska@umw.edu.pl (M.K.); aleksandra.mikolajczyk@umw.edu.pl (A.M.)

**Keywords:** antimicrobial peptides, halophiles, metagenomics, molecular docking, sequencing

## Abstract

Microorganisms inhabiting saline environments have been known for decades as producers of many valuable bioproducts. These substances include antimicrobial peptides (AMPs), the most recognizable of which are halocins produced by halophilic Archaea. As agents with a different modes of action from that of most conventionally used antibiotics, usually associated with an increase in the permeability of the cell membrane as a result of a formation of channels and pores, AMPs are a currently promising object of research focused on the investigation of antibiotics with non-standard modes of action. The aim of this study was to investigate antimicrobial activity against multidrug-resistant human pathogens of three peptides, which were synthetised based on sequences identified in metagenomes from saline environments. The investigations were performed against *Enterococcus faecalis*, *Staphylococcus aureus*, *Klebsiella pneumoniae*, *Acinetobacter baumannii*, *Pseudomonas aeruginosa*, *Escherichia coli* and *Candida albicans*. Subsequently, the cytotoxicity and haemolytic properties of the tested peptides were verified. An in silico analysis of the interaction of the tested peptides with molecular targets for reference antibiotics was also carried out in order to verify whether or not they can act in a similar way. The P1 peptide manifested the growth inhibition of *E. faecalis* at a MIC_50_ of 32 µg/mL and the P3 peptide at a MIC_50_ of 32 µg/mL was shown to inhibit the growth of both *E. faecalis* and *S. aureus*. Furthermore, the P1 and P3 peptides were shown to have no cytotoxic or haemolytic activity against human cells.

## 1. Introduction

Microorganisms inhabiting high-salinity environments are known as halophiles. Representatives of these class of extremophiles can be found in all three domains of life [1]. Halophiles are known as a source of active biomolecules for biotechnological and pharmaceutical applications and due to the fact of high variability and unique adaptation to extreme environments they are still interesting subjects in the search for novel bioproducts useful in biomedicine and industry. In the era of multidrug-resistant bacteria, an increasing morbidity rate of cancer, and extreme environmental pollution, research on the development of new active compounds is urgently necessary [2,3].

Antibiotic resistance is one of the most significant public health problems in recent years. Due to the continuous overuse of antibiotics, both in human medicine and in other applications, the number of cases of infections with pathogens resistant to standard antibiotics is increasing year by year [4]. In order to combat this threat, the World Health Organization introduced the “Global Action Plan on Antimicrobial Resistance” in 2015, where the guidelines for global actions to prevent the further development of antibiotic resistance was presented. One of the important activities under this plan is an exploration of the sources and mechanisms of antibiotic resistance that are created in the environment in accordance with the One Health approach [5]. In addition, the key activity also refers to the seeking of new active antimicrobial compounds, in particular antibiotics with new mechanisms of action. Furthermore, antimicrobial peptides (AMPs) can be included in this group of potential biomolecules to cope with microbials resistance, respectively. Comparable activities have been undertaken by the European Union since 2003 through public health programs. Here, supporting actions for research activities and programs in order to search for new antimicrobial drugs, innovative products and alternative methods of therapy, as well as the prevention of infections and infectious diseases have been underlined. Moreover, the European Union Joint Action on Antimicrobial Resistance and Healthcare Associated Infections has been established to support European Union members in developing and implementing One Health solutions to combat the growing threat of antibiotic resistance.

Antimicrobial peptides (AMPs), a diverse group of bioactive small proteins, are a part of the body’s first line of defence for pathogen inactivation in eucaryotic organisms. They work by disrupting bacterial cell membranes, modulating the immune response, and regulating inflammation [6]. AMPs serving as important tools of microorganism competition in complex microbial communities can inhibit the growth of other species of microorganisms as well [7,8]. AMPs that are valuable and important for biotechnology and medicine were identified in a number of eukaryotic organisms and numerous bacteria and archaea. However, there are also plenty of unconventional sources of AMPs, including unculturable soil and marine bacteria, extremophilic microorganisms and methods available to produce vast libraries of derivatives [9,10,11]. Currently, only few AMPs are approved for clinical usage as alternatives to antibiotics in terms of their antimicrobial potency. This behaviour is related to the challenges to the clinical application of AMPs which include cytotoxic effects, costs of production and obstacles related to peptide bioavailability and efficacy [12]. However, in the age of antibiotic resistance, AMPs have the potential to be a valuable tool with which to combat multidrug resistant bacteria and reduce usage of antibiotics in clinical applications due to the synergistic activity of AMPs with antibiotics [13,14]. AMPs can be used not only in microbial infection treatment, but also can be applied in agriculture to control plant diseases, as therapeutics in aquaculture or food additives for livestock [15,16,17]. Finally, AMP usage as food preservatives is also proven. Nisin is an AMP representative as for the antimicrobial active packaging of food compounds [18,19].

Microorganisms inhabiting saline and hypersaline environments are also producers of wide range of AMPs. Halophilic archaea produce group of AMPs known as halocins [10]. First discovered halocin, Halocin H4, was described by Francisco Rodriguez-Valera in 1982 [20]. Only for Halocin H4, C8 and S8 are the amino acid and nucleotide sequences of their genes known. Presented halocins are synthesised in the form of precursor proteins which undergo post-translational modification to release mature halocins as secretory proteins [10]. Additionally, halophilic bacteria are AMP producers, but they are less-explored than are archaea [2,21]. Despite the ecological and environmental role of several halocins, their action against human pathogens has been less-studied. The clinical significance of AMPs produced by halophilic microorganisms is limitedly reported and their antimicrobial activity against the most important human pathogens still remains an area of preliminary research [2]. However, halocin stability in hypersaline environments made them useful in industrial applications especially in the preservation of salted food [10,21].

The aim of this study was to investigate antibacterial activity against multidrug-resistant human pathogens: *Enterococcus faecalis*, *Staphylococcus aureus*, *Klebsiella pneumoniae*, *Acinetobacter baumannii*, *Pseudomonas aeruginosa*, *Escherichia coli* and *Candida albicans* of three peptide sequences identified from metagenomes of the Bochnia Salt Mine and brine graduation towers. The activity of the selected peptides in combination with the reference antibiotics was also checked. To verify the cytotoxicity and potential haemolytic effect, specific assays were also performed. The potential mechanism of action for the peptides with confirmed antibacterial activity was studied via in silico analysis. 

## 2. Results

### 2.1. AMPs Sequences Identification

Eight metagenomes from high-saline environments were analysed to identify the AMP sequences with potentially the best antimicrobial properties. In assembled contigs from all eight samples, 271 potentially AMP sequences were identified. Briefly, 191 sequences were marked as possible haemolytic activity, which resulted in their exclusion from further analysis. From the 80 remaining sequences, peptides with a probability of being an AMP of above 70% were selected. The results predicted only three sequences; thus they have been chosen for synthesis and the in vitro evaluation of their antimicrobial properties.

The physico-chemical properties such as the net charge, molecular weight, and hydrophobicity of the peptides selected for in vitro analysis have been determined in order to evaluate their essential properties. All peptides are characterised by a positive net charge and contain significant number of hydrophobic amino acids which may indicate their amphipathic properties. Described properties and AMP prediction results are presented in Table 1.

In Table 2, the pathogens and the reference antibiotics which are used in the treatment of bacterial infection are presented.

The receptors listed in Table 2 were also used in docking for peptides P1, P2, and P3. *E. faecalis* Topoisomerase IV, *S. aureus* DNA gyrase and *E. coli* rRNA 16S, were selected as receptors for molecular docking due to the mode of action of levofloxacin and gentamicin. Gentamicin is supposed to incorporate and destabilise the bacterial outer membrane and reach the 16S rRNA where it binds in helix 44 near the A-site of the 30S ribosomal subunit [22]. Levofloxacin inhibits two bacterial enzymes: DNA gyrase and Topoisomerase IV. Both targets are type II topoisomerases but have unique functions within the bacterial cell [23].

### 2.2. Antimicrobial Activity of Selected Peptides

An antimicrobial activity assay was performed using multidrug-resistant pathogens. First, the MICs for antibiotics used as reference agents were verified. The obtained MIC value for *S. aureus* 43300 was levofloxacin 0.25 µg/mL, that for *E. coli* 25922 was gentamicin 4 µg/mL, that for *C. albicans* 10231 was amphotericin B 1 µg/mL, that for *P. aeruginosa* 27853 was levofloxacin 1 µg/mL, that for *E. faecalis* 29212 was levofloxacin 1 µg/mL, that for *A. baumannii* 19606 was levofloxacin 0.5 µg/mL and that for *K. pneumoniae* 700603 was gentamicin 4 µg/mL. 

The results for P1, P2 and P3 did not present full growth inhibition for any of the strains. Nevertheless, the MIC_50_ was achieved for P1 and P3 against the selected MDR strains. The assay showed a MIC_50_ of 32 µg/mL for P1 and P3 against *E. faecalis*. Moreover, P3 inhibited the growth of MRSA with a MIC_50_ of 32 µg/mL and that of *E. coli* with a MIC_50_ of 256 µg/mL. P2 did not inhibit the growth of any of the tested microorganisms (Table 3). 

### 2.3. Antimicrobial Activity of Peptides in Combination with Levofloxacin against E. faecalis and MRSA

After MIC_50_ detection, peptides P1 and P3 were tested in combination with levofloxacin against *E. faecalis* and *MRSA* using a microdilution method and checkerboard assay. The checkerboard assay was performed for potential synergy effect identification. Levofloxacin was selected for this study as the reference antibiotic for all strains, where MIC_50_ was defined. For the microdilution method, the MIC value against *E. faecalis* for P1 and P3 was obtained with a combination of 0.03 µg/mL of levofloxacin and 32 µg/mL of each peptide, respectively. In comparison, levofloxacin alone showed an inhibitory effect (MIC) at a concentration of 1 µg/mL. The MBC for levofloxacin alone and for its combination with peptides was the same and it was 1 µg/mL. 

In the case of *S. aureus* 43300, the MIC for the combination of levofloxacin and a peptide was determined to be 0.03 µg/mL of levofloxacin and 32 µg/mL of peptide P3. Levofloxacin alone presented a MIC of 0.25 µg/mL. Furthermore, 0.25 µg/mL of levofloxacin in combination with 32 µg/mL of peptide P3 was essential for MBC determination.

For checkerboard assay results, the FIC index could not be calculated, because the MIC value was not determined in a single treatment of P1 and P3. Nevertheless, the checkerboard test confirmed the experimental values of the microdilution method. Here, inhibition of growth for *E. faecalis* and *MRSA* was also obtained (that of levofloxacin for *E. faecalis* was 4/2/1 µg/mL and that for *MRSA* was 1/0.5/0.25 µg/mL). The MIC remained the same compared to levofloxacin alone. Moreover, for P1 and P3 in the concentration of 32 µg/mL only MIC_50_ was determined and confirmed. For all other concentration combinations, where the concentration of levofloxacin was lower than the MIC and the AMP concentration was lower than 32 µg/mL, no antimicrobial activity was determined. This may suggest that no synergy effect can be obtained for analysed systems. 

### 2.4. Cytotoxicity and Haemolytic Properties

For the P1 and P3 peptides, cytotoxic properties were also investigated using the Neutral Red assay. The experiment was performed for P1, P1–levofloxacin (1 µg/mL), P3, and P3–levofloxacin (1 µg/mL) in three-point time frame (24/48/72 h). Viability calculation for all systems in each time estimation gave positive results. The viability range was 100–107% after 24 h, 100–115% after 48 h and 99–107% after 72 h. Figure 1 presents the results for all systems after 72 h of incubation. 

In the second assay, where potential haemolytic activity was checked, the analysed peptides did not show cytotoxicity towards human erythrocytes. The level of haemolysis for the peptide-treated red blood cells was similar to the level of haemolysis in the untreated samples (Table 4).

### 2.5. Molecular Docking Analysis

#### 2.5.1. Molecular Docking Results between *E. faecalis* 29212 Topoisomerase IV and Levofloxacin as well as *E. faecalis* 29212 Topoisomerase IV and Peptides P1 and P3

Docking studies were performed in order to estimate the binding interactions of antibiotics and compare the results with those of the studied peptides: P1, P2, and P3. Docking studies revealed that levofloxacin showed better binding affinity (−7.7 kcal/mol) than did peptide P1 (−7.0 kcal/mol) and P3 (−6.6 kcal/mol). The amino acids involved in the formation of hydrogen bonds with Topoisomerase IV are presented in Table 5. The visualisation of the interaction of the peptides with Topoisomerase IV is presented in the form of 3D models in Figure 2 and Figure 3.

#### 2.5.2. Molecular Docking Results between *S. aureus* 43300 DNA Gyrase and Levofloxacin as well as *S. aureus* 43300 DNA Gyrase and Peptide P3

Levofloxacin was found to form a hydrogen bond with Asn383 with *S. aureus* 43300 DNA gyrase. A binding affinity of −10.2 kcal/mol was found for the best-ranked pose. Peptide P3 made hydrogen bonds with Glu436, Arg433, Ser31, Asn535, Asp182, Lys59 and Arg61. The binding affinity of the interaction was −8.8 kcal/mol. The visualisation of the interaction of the peptide P3 with DNA gyrase is presented in the form of 3D model in Figure 4.

#### 2.5.3. Molecular Docking Results between *E. coli* 25922 16S rRNA and Gentamicin as well as *E. coli* 25922 16S rRNA and Peptide P3

Gentamicin C1a binds in the A-site of 16S rRNA and forms hydrogen bonds to four nucleotides: G1405, U1495, A1408 and A1492. A binding affinity of −7.8 kcal/mol was found for the best-ranked pose. Peptide P3 made hydrogen bonds in the A-site rRNA (binding affinity −3.3 kcal/mol) with G1405, U1495, G1494, A1493, A1492, and G1491. The visualisation of the interaction of the peptide P3 with A-site of 16S rRNA is presented in the form of 3D model in Figure 5.

## 3. Discussion

Due to the challenging fact of increased antibiotic resistance, the crucial task is to seek for alternatives in pathogen defence. Multidrug resistance is extremely hard to neglect and may be untreatable with conventional antibiotics [24]. Antimicrobial peptides—AMPs—can be an alternative with which to overcome bacterial resistance. In our study, the sequences of 271 AMPs were identified in metagenomic data from saline environments and three of them with the highest probability of being AMP were selected for further analysis. Their antimicrobial properties, cytotoxic and potential haemolytic effects against eukaryotic cells and potential mechanism of action were characterised. Due to the characteristics of the environmental niche of microorganisms in the metagenome of which genes encoding the analysed AMPs were detected, it can be assumed that they may belong to halocins. Unlike the majority of studies that assess AMPs produced by halophilic microorganisms, we decided for ours to focus on assessing their activity against pathogens characterised by increasing antibiotic resistance [10]. Most of the known AMPs isolated from halophilic microorganisms were tested or showed antimicrobial activity only against closely related species. The production of AMPs active against closely related strains indicates their important role in competition between strains inhabiting the same niches [20,25,26]. However, when it comes to the application of AMPs produced by halophiles against pathogenic strains, only few examples can be found. Halocin KPS1 showed a broad spectrum of antimicrobial activity, including pathogenic strains such as *Streptococcus mutans* MTCC896, *Bacillus subtilis* MTCC1134, *Escherichia coli* MTCC1671, *Staphylococcus aureus* MTCC916, and *Pseudomonas aeruginosa* MTCC6538 [27]. The cell-free supernatant of the halocin-producing strain *Halobacterium salinarum* ETD5 has also been shown to inhibit the growth of *P. aeruginosa*. However, it has not yet been possible to isolate the halocin responsible for this effect [28,29]. 

Due to their activity against *E. faecalis* and *S. aureus* and their lack of cytotoxicity and haemolytic activity, AMPs P1 and P3 may be interesting candidates for clinical use. They can also be applied as a starting point for further modifications and the creation of semi-synthetic AMPs with preferred properties. Particularly promising seems to be P3, which induced the growth inhibition of both previously indicated strains at a MIC_50_ of 32 µg/mL. P1 caused growth inhibition only for *E. faecalis*. Comparing the MIC_50_ obtained for our AMPs to the AMPs tested in clinical and preclinical trials, peptides such as HB1275, HB1345 or LTX-109 showed inhibitory activity at concentrations in the range of 1–4 µg/mL [30]. On the other hand, there are also examples of AMPs that are active in a similar or higher range of concentrations. An example here is the MSI-78 peptide, which has a MIC_50_ of 31.3 µg/mL for *E. faecalis* and >125 µg/mL for MRSA. It is worth noting that the MSI-78 peptide has reached the third stage of clinical trials and is treated as a promising basis for further modifications to improve its therapeutic properties [31]. Another example may be Peptidolipin B-F for which the MIC_50_ for MRSA is 64 µg/mL [32]. Combinations of the test peptides with reference antibiotics were also tested to investigate their possible interactions. However, unlike some AMPs, the P1 and P3 peptides did not show any beneficial interactions with the reference antibiotics [33].

In terms of length, molecular weight and net charge, our AMPs fall within the range of other AMPs tested in clinical and preclinical trials. Most of the AMPs in this group are rather small and their length and molecular weight are less than 25 amino acids and 3100 Da. With a molecular weight of 3575 Da and 4298 Da, our peptides can be classified as quite large. However, there are also larger peptides, such as LL-37 (4490 Da), AP114 (4417 Da) and AP138 (4460 Da). In addition, the net charge of our peptides is almost in the middle of the range for clinically tested AMPs, which extends from −3 to 14 [30].

Molecular modelling techniques are being used in the pharmaceutical industry as they are an effective tool in the drug discovery processes. Most of the drugs produce their effect by interacting with a biological macromolecule such as an enzyme, DNA or a receptor. Discovering and developing any new medicine is a long and expensive process. A new compound must have the ability to replace the existing cure with a desired response and with minimal side effects [34]. Molecular docking is used to model the interaction between compound and target protein receptors. 

An estimation of the binding pose demonstrates possible interactions [35]. The docking of substances with known action modes such as levofloxacin and gentamicin was used as a starting point to compare the interactions of the studied proteins. Levofloxacin is bactericidal and exerts its antimicrobial effects via the inhibition of bacterial DNA replication. Levofloxacin exerts its antimicrobial activity via the inhibition of two key bacterial enzymes: DNA gyrase and Topoisomerase IV [36]. Gentamicin is effective against both gram-positive and gram-negative organisms but is particularly useful for the treatment of severe gram-negative infections. Structural and cell biological studies suggest that it binds to the 16S rRNA [37]. Due to the published mechanism of action for levofloxacin and gentamicin Topoisomerase IV of *E. faecalis*, the DNA gyrase of *S. aureus*, and 16S rRNA of *E. coli* were selected as target receptors for in silico analyses. The docking of levofloxacin and gentamicin was performed to compare the possible mechanisms of action and potential hydrogen bonds formed in order to stabilize the docked peptide. Kiranpreet Kaur et. al. in their analysis also compared the docking of novel modified AMPs based on AMPs with known functions and mechanisms of action to find out the best conformation of a compound [38]. In the study conducted by Abdulmalik Aliyu et al., AMPs were designed, and molecular docking was used to compare the antimicrobial activity of novel AMPs with that of the reference [39]. Roy A. et. al. investigated the inhibition of topoisomerases by synthetic peptides. Synthetic peptides have been proposed to be the ideal inhibitors of enzyme activity either alone or in combination with small-molecule drugs [40]. Chandrashekar S. et al. conducted a study, where the protein NAP1 inhibited a *S. aureus* gyrase–AM8191 complex [41]. Molecular modelling study revealed that peptide P1 and P3 interacted with Topoisomerase IV of *E. faecalis* and formed hydrogen bonds with Ans650 as a reference antibiotic used in the treatment—levofloxacin. Peptides P1 and P3 require additional in-depth analyses. The molecular docking of peptides P1 and P3 revealed that the peptides formed hydrogen bonds to different residues from those of the reference antibiotics in the case of *E. coli* and *S. aureus*. 

According to Zhang et al., AMPs may act as a novel therapeutic option for treating antibiotic-resistant bacteria either alone or applied in a synergistic manner with antibiotics. Penicillin or chloramphenicol combined with nisin improved the antibacterial effect in *E. faecalis* where single antibiotic alone had no significant activity [42]. In addition, due to the fact that the tested peptides come from strains living in saline environments, it is quite likely that they are characterised by high tolerance to high salinity. This creates the opportunity not only to use the tested peptides in medicine, but also to use them in industry, for example for food preservation [18,19]. The relatively high MIC_50_ for the tested peptides can be considered acceptable due to the lack of toxicity at these concentrations. Of course, it is beneficial to conduct further research aimed at increasing the activity of the tested peptides while maintaining their low toxicity. Such actions may include modifications of the AMP sequence or chemical modifications such as lipidation, glycosylation, guanidination, hydrazidation and small molecule conjugation [43].

In conclusion, as a result of the analysis of metagenomic sequences from environmental samples from saline environments, three peptide sequences with a high probability of having antimicrobial activity were selected. On the basis of these sequences, peptides with physico-chemical parameters typical of AMP were synthesised. The laboratory evaluation showed that the P1 and P3 peptides have antimicrobial activity. The P1 peptide inhibited the growth of *E. faecalis* and the P3 peptide inhibited the growth of the *E. faecalis* and *S. aureus* strains. It has also been shown that these peptides have no cytotoxic or haemolytic effect. As a part of the in silico analyses, the possible mode of action of the peptides was demonstrated, through pathways analogous to those of the reference antibiotics. However, it should be borne in mind that the presented variant of the mechanism of action is one of many options, and determining the proper mode of action for each of the peptides requires in-depth research in this area. The results obtained from all tests performed indicate that the P1 and P3 peptides may be valuable candidates for the role of antimicrobial agents.

## 4. Materials and Methods

### 4.1. Sample Collection and Sequencing

Four brine samples were collected from brine wells situated in the Bochnia Salt Mine located in southern part of Carpathian Foreland in Poland near Kraków city (49°58′09″ N 20°25′03″ E), three samples of blackthorn were collected from three brine graduation towers located in Lodz city in Podolski Park (51°74′14″ N 19°49′17″ E), Botanik Residential (51°75′02″ N 19°40′30″ E), and Mikolaj Rej Residential (51°78′83″ N 19°41′73″ E) and one sample of brine was collected from the Tadeusz brine source in Zablocie (49°90′77″ N 18°77′06″ E). All samples were collected in 2019. To prepare the brines for DNA isolation, 2 mL of saline was transferred into a new sterile Eppendorf tube and centrifuged for 10 min at 14,000× *g*. After centrifugation, 1.8 mL of brine was extracted, and 1.8 mL of new brine was added. The centrifugation was then repeated. This procedure was repeated 5 times for each sample. In the case of blackthorn, about 1 cm fragments of twigs were rinsed and shaken in sterile nuclease-free water and then the water was used for DNA isolation. DNA was extracted from the samples using PowerSoil DNA Isolation Kit (Mo Bio Laboratories Inc., Carlsbad, CA, USA). Samples were eluted with nuclease-free water in a volume of 50 µL. DNA concentration was determined using the Qubit high-sensitivity (HS) assay kit (ThermoFisher, Waltham, MA, USA).

Shotgun sequencing libraries were prepared using Vazyme TruePrep DNA Library Prep Kit V2 for Illumina (Vazyme, Nanjing, China). Libraries were sequenced on Illumina NextSeq 500 (Illumina, San Diego, CA, USA) with 2 × 150 bp paired-end reads. 

### 4.2. In Silico Analysis of Selected Organisms

For shotgun sequencing, data de novo assembly was performed using MEGAHIT v.1.2.9 with the “--min-contig-len 1000” parameter [44]. The quality of the assemblies was checked using metaQUAST [45]. 

Identification of AMP sequences was performed using marcel v. 0.3.1 with default parameters [46]. AMP sequences with non-haemolytic properties were verified and classified into functional types using the iAMP-2L platform [47]. 

In assembled contigs, potentially AMP sequences were identified. Sequences marked as a potentially haemolytic were excluded from further analysis. The probability of non-haemolytic AMP sequence classification ranged from 50.05% to 76.2%. Furthermore, sequences where the AMP probability exceeded 70% were selected for the synthesis and in vitro evaluation of their antimicrobial and cytotoxic properties. Peptides labelled as P1 and P3 were supposed to affect bacteria and P2 fungi. 

The possible mode of action and target organisms were assumed using CAMP database [48]. CAMPR4 contains information on the AMP sequence, protein definitions, accession numbers, activity, source organisms, target organisms, protein family descriptions, N and C terminal modifications and links to databases such as UniProt, PubMed and other antimicrobial peptide databases.

Peptide parameters such as total net charge, molecular weight, hydrophobicity were calculated using the APD3 database [49].

### 4.3. Antimicrobial Peptide Synthesis and Preparation

Three peptides with the highest AMP probability were synthesised by Pepmic Co., Ltd. (Suzhou, China). At least 98% purity was required of each peptide. Peptides were synthesised and ligated at pH~7 with a thiophenol derivative. Quality analysis of peptides was conducted using the MS spectrum. The QC certificate was delivered by the provider. Additional QC checking was performed using the ESI-MS mass spectrometer by Bruker Daltonik (model Compact, Billerica, MA, USA).

### 4.4. Antimicrobial Activity Assay

Seven reference strains from the ATCC collection (*A. baumanii* 19606, *K. pneumoniae* 700603, *S. aureus* 43300, *E. coli* 25922, *E. faecalis* 29212, *P. aeruginosa* 27853 and *C. albicans* 10231) were used for the antimicrobial activity assay. The antimicrobial activity assay was performed in accordance with the standard protocol using the microdilution method with spectrophotometric measurements (λ = 580 nm at the starting point and after 24 h) [50] adhering to the ISO standard 20776-1:2019 [51], ISO standard 16256:2012 [52] and modified Richard’s method [53,54,55]. 

Stock peptide solutions were prepared in water. Serial dilutions were made on 96-well microplates in the range between 0.5 µg/mL and 256 µg/mL. Tryptone soy agar (TSA) plates were inoculated with microbial strains from the studied stocks. After 24 h/37 °C of incubation (for bacteria) or 24 h/25 °C of incubation (for fungus), a proper density of the bacterial and fungal suspension was prepared using a densitometer (final inoculum (5 × 10^5^ CFU/mL) was prepared in tryptic soy broth (TSB). A positive (TSB + strain) and negative control (TSB) were also included in the test. Spectrophotometric solubility control of each peptide was also performed. Microplates were incubated at 37 °C or 25 °C for 24 h on the shaker (500 rpm). After this, the spectrophotometric measurement was performed at 580 nm and then 50 µL aliquots of 1% (m/v) 2,3,5-triphenyltetrazolium chloride (TTC) solution were added into each well. TTC is a chemical indicator which is converted into red formazan crystals in living microbial cells. A possible killing effect can be observed as the lowest concentration determined via visual analysis after 24 h of incubation with TTC (which did not change the colour to pink) Thus, the simultaneous usage of the microdilution method and the TTC examination let us to determine the potential MIC (minimal inhibitory concentration) and MBC (minimal bactericidal concentration) or MFC (minimal fungicidal concentration) values. 

For each strain, the validation process was performed using the following antibacterial/antifungal agents: levofloxacin, gentamicin, and amphotericin B, all used in accordance with the EUCAST examination. The minimal inhibitory concentration (MIC) was determined for each strain exposed to antibacterial/antifungal agents. Moreover, for the peptide investigation, the 50% minimal inhibitory concentration (MIC_50_) was determined. The original aim of the study was to determine the MIC for each of the tested compounds, including peptides, but due to the inability to determine the MIC of peptides in the tested concentration range, it was decided to determine the MIC_50_. As MIC_50_ is the lowest concentration of an antimicrobial agent that inhibits the measured microbial growth to 50%, as referred to previously, a positive control was obtained. 

After the MIC_50_ determination for P1 (*E. faecalis*) and P3 (*E. faecalis*, *S. aureus*), a synergy checkerboard assay was performed. Columns 1 to 11 contain 2-fold serial dilutions of P1 or P3 which start from earlier-obtained MIC_50_ values, and rows A to G contain 2-fold serial dilutions of levofloxacin. Levofloxacin was selected for this study as the reference antibiotic for tested strains. Column 12 contains a serial dilution of levofloxacin alone, while row H contains a serial dilution of Compound A alone. A detailed scheme of plate preparation for the checkerboard assay is presented at Figure 6.

### 4.5. Neutral Red Cytotoxicity of Selected AMPs

For each peptide, where the antimicrobial activity was determined against *E. faecalis* and *MRSA* strains, a Neutral Red (NR) cytotoxicity assay was performed using human primary renal proximal tubule epithelial cells (RPTEC) from the ECACC collection. The experiment was performed in accordance with ISO:10993 guidelines (Biological evaluation of medical devices; Part 5: Tests for in vitro cytotoxicity; Part 12: Biological evaluation of medical devices, sample preparation and reference materials: ISO 10993-5:2009 and ISO/IEC 17025:2005). A standard protocol for the NR assay was used from Nature Protocol [56]. MEMα supplemented with 10% FBS, 2 mM L-glutamine and a suitable amount of antibiotics (amphotericin B, gentamycin) were used for the experiment. Stock peptide solutions were prepared in water and then diluted 100× in the medium. The P1 and P3 tested concentrations were in a range from 64 to 8 µg/mL. After adding the proper mixtures of the testing compounds and cells (1 × 10^5^ cells/mL) into each well, plates were incubated for 24, 48 and 72 h in 5% CO_2_ at 37 °C. Next, the medium was removed and 100 µL of NR solution (40 µg/mL) was added to each well followed by incubation for 2 h at 37 °C. After removing the dye, the wells were rinsed with PBS and left to dry. Then, the NR solution (1% of glacial acetic acid, 50% of 96% ethanol and 49% of deionised water; *v*/*v*) was added to each well. The plates were shaken (30 min, 500 rpm) until NR was extracted from the cells and formed a homogenous solution. The absorbance was measured using a microplate reader at λ = 540 nm. As a negative control, untreated cells were considered to have 100% of the potential cellular growth. Furthermore, cells incubated with 1 µM staurosporine were used as a positive control.

### 4.6. Haemolytic Activity of AMPs

Haemolytic activity was determined by incubating a 5% (*v*/*v*) suspension of human erythrocytes with selected peptides. Red blood cells were rinsed three times in PBS, via centrifugation for 15 min at 1500× *g*. Then, they were incubated at 37 °C for 3 h with saponin (positive control), with additional PBS (negative control) and with specific peptides of the MIC_50_. Afterwards, the samples were centrifuged at 1500× *g* for 15 min, the supernatant was separated from the pellet and the absorbance of the supernatant was measured at λ = 540 nm. Pure PBS was used as a blank control. The relative haemolysis percentage (H) was computed using the following equation:$$H=\frac{absorbance\;of\;sample-absorbance\;of\;blank}{absorbance\;of\;positive\;control-absorbance\;of\;blank}\times 100$$

### 4.7. Molecular Docking

Structures of selected peptide sequences were predicted using the Psipred database [57] and modelled using RoseTTAFold provided by computing resources of the Baker lab. Peptides P1 and P3 were found to have antimicrobial activity against three pathogens out of the seven used in laboratory tests (*E. faecalis* 29212, *S. aureus* 43300 and *E. coli* 25922). Molecular docking was performed to model the interaction between molecules and a receptor at the atomic level, in order to characterise the behaviour of molecules in the binding site of a target proteins as well as to elucidate fundamental biochemical processes. 

Molecular docking was performed using AutodockVina [58] in order to dock levofloxacin and gentamicin, and ClusPro [59] server was used for the docking of peptides P1, P2, and P3. Input sequences were prepared using the AutoDockTools software. 

The structures used in docking were optimised via energy minimisation in the MMFF94 force field [60]. Water molecules were removed from receptors and polar hydrogens were added, while missing atoms were repaired. ADT 1.5.6 software [61] was used to investigate activity in terms of binding affinity (Kcal/mol). The docking outcomes, e.g., bonds between ligand and receptor and the binding affinity score for best-docked conformation, were compared for the reference antibiotics and analysed peptides and presented in the Results section. 

## Figures and Tables

**Figure 1 ijms-24-11787-f001:**
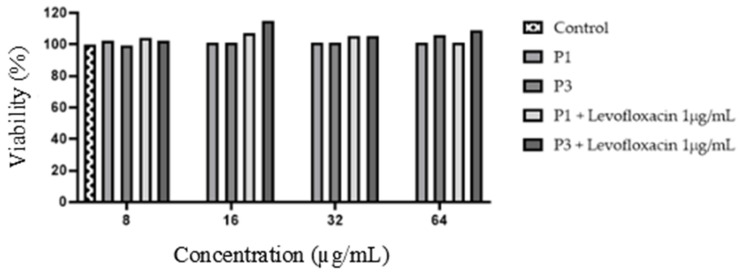
RPTEC viability after 72 h of incubation with P1/P3 peptides alone and in combination with levofloxacin.

**Figure 2 ijms-24-11787-f002:**
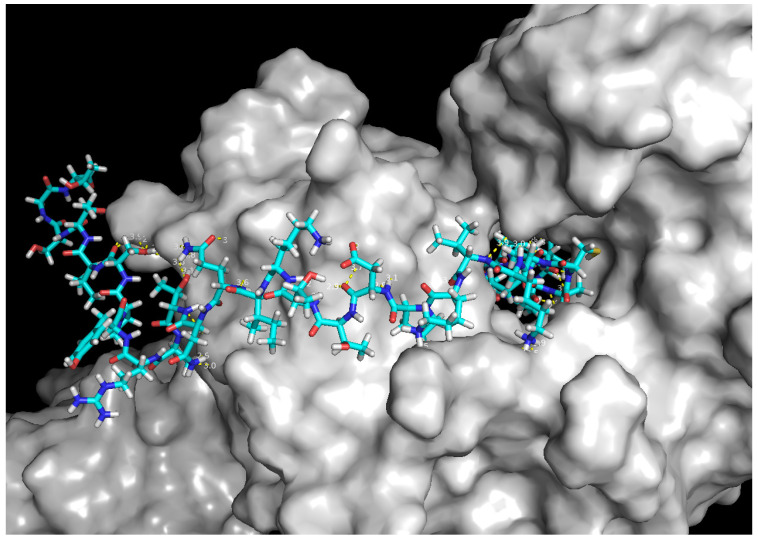
3D model of peptide P1 interaction with Topoisomerase IV.

**Figure 3 ijms-24-11787-f003:**
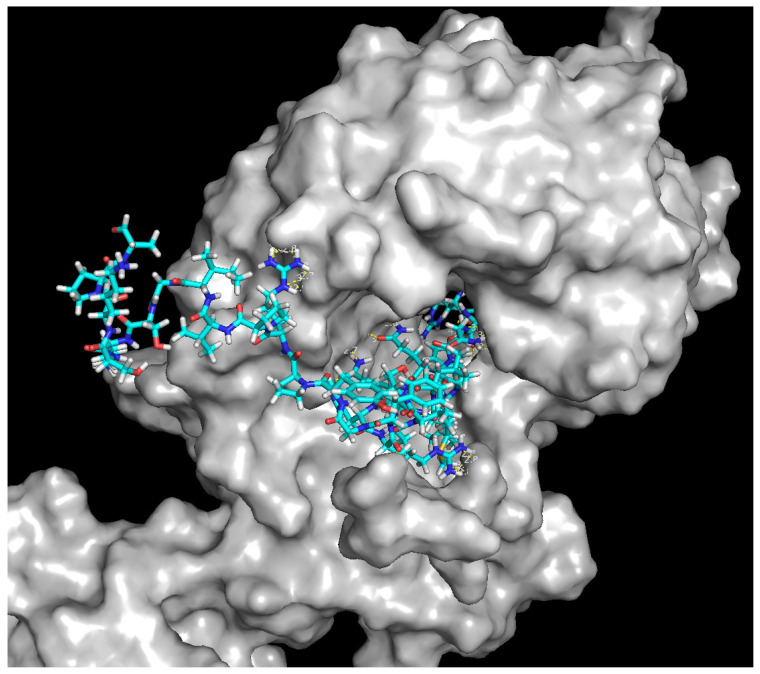
3D model of peptide P3 interaction with Topoisomerase IV.

**Figure 4 ijms-24-11787-f004:**
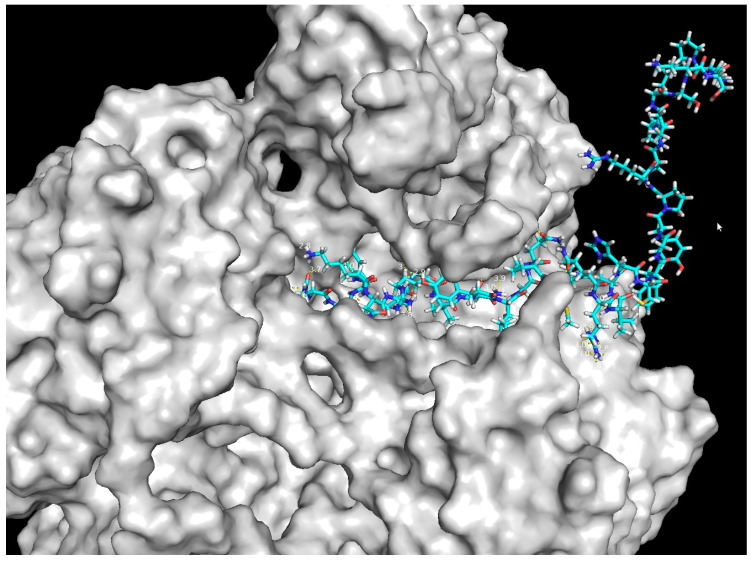
3D model of peptide P3 interaction with DNA gyrase.

**Figure 5 ijms-24-11787-f005:**
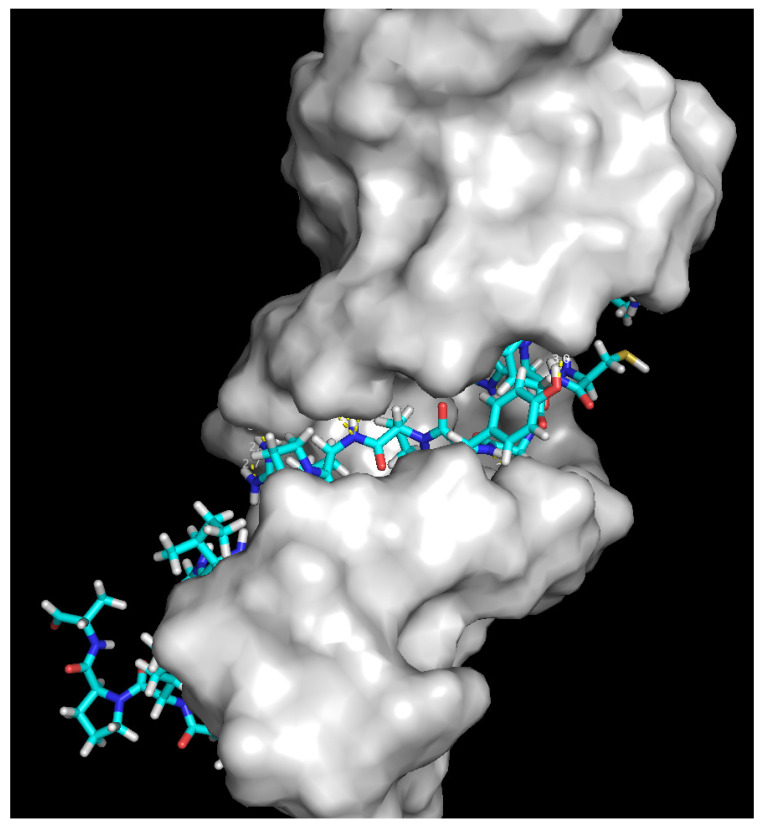
3D model of peptide P3 interaction with A-site of 16S rRNA.

**Figure 6 ijms-24-11787-f006:**
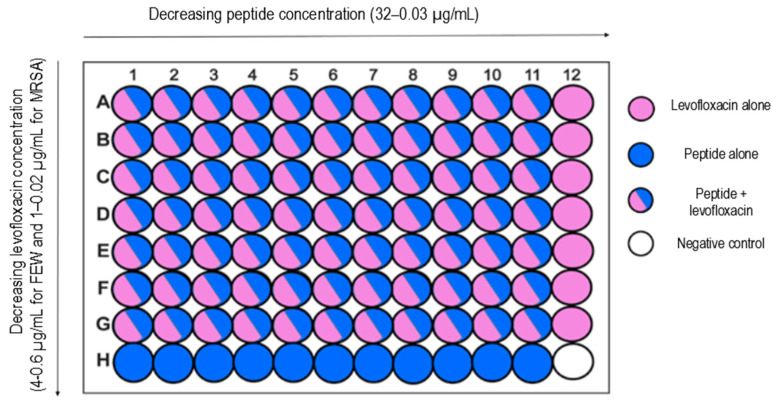
Synergy checkerboard assay plate scheme.

**Table 1 ijms-24-11787-t001:** Physicochemical properties of peptides selected for antimicrobial activity testing.

ID	GenBank Accession	Source	Sequence	Number of Residues/Total Net Charge	Hydrophobic Amino Acids	Molecular Weight (g)	AMP Probability	Target Organism
P1	OR326674	Brine graduation tower—Botanik Residential	LAAIDALARACLKVKPDTTKIQNTARYPSVTSGT	34 (Total net charge + 3)	13/34 (38%)	3575.142	76.2%	Bacteria
P2	OR326675	Brine well—The Bochnia Salt Mine	AAALCVRAAVFKRGESNGYDPKPGDLRVGKVKRAERRVEAC	41 (Total net charge + 5)	17/41 (41%)	4445.154	72.3%	Fungi
P3	OR326676	Brine graduation tower—Podolski Park	NHFKNIGRVNYLGQPMLQRVSHCFGYPRPVIGSKSKPA	38 (Total net charge + 6.5)	12/38 (32%)	4298.023	71.3%	Bacteria

**Table 2 ijms-24-11787-t002:** Pathogens used in molecular docking with reference antibiotics and their molecular targets.

Strain	Reference Antibiotic	Receptor
*E. faecalis* 29212	Levofloxacin	Topoisomerase IV
*S. aureus* 43300	Levofloxacin	DNA gyrase
*E. coli* 25922	Gentamicin	16S rRNA

**Table 3 ijms-24-11787-t003:** MIC_50_ (µg/mL) of peptides against selected microbials.

Strain	P1	P2	P3	Reference Antibiotic (MIC µg/mL)
*A. baumannii* 19606	>256	>256	>256	0.5
*C. albicans* 10231	>256	>256	>256	1
*E. coli* 25922	>256	>256	256	4
*E. faecalis* 29212	32	>256	32	1
*K. pneumoniae* 700603	>256	>256	>256	4
*S. aureus* 43300 (MRSA)	>256	>256	32	0.25
*P. aeruginosa* 27853	>256	>256	>256	1

**Table 4 ijms-24-11787-t004:** Haemolytic activity of tested peptides.

Treatment Group	Mean Percentage Haemolysis (±SEM)
Untreated cells	7.53 ± 0.61
Saponin	100 ± 5.74
P1	6.81 ± 0.45
P2	7.65 ± 0.10
P3	7.29 ± 0.43

**Table 5 ijms-24-11787-t005:** Amino acids involved in the formation of hydrogen bonds with Topoisomerase IV.

Agent	Hydrogen Bonds
Levofloxacin	Asn650
P1	His511, Glu512, Tyr514, Lys534, Leu540, Arg572, Glu575, Ile576, Glu578, Ile589, Ile594, Val596, Glu647, Thr649, Asn650, Asn702 and Glu807
P3	Leu540, Asp590, Tyr597, Lys606, Lys613, Gln645, Asn650, Val651, Leu653, Asp656, Asp658, Asn702, Glu805, Glu808 and Glu815

## Data Availability

Data will be made available on request.

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
