# Peer review of "Novel Antimicrobial Peptides from Saline Environments Active against E. faecalis and S. aureus: Identification, Characterisation and Potential Usage"

_ijms, 2023, doi:10.3390/ijms241411787_

Round 1

Reviewer 1 Report

Please see comments in the provided file.

Please see comments in the provided file.

Author Response

Thank you very much for your comments and for reviewing our manuscript. We hope that the introduced changes will be satisfactory. Responses to comments are included in the attached file.

Reviewer 2 Report

The authors of this study identified three peptide sequences from metagenomes of saline environments that have potential as antimicrobial peptides (AMPs). Despite their relatively lower potency and lack of synergy with antibiotics, the identification of these peptides from extreme saline conditions is a significant finding that warrants publication. However, the manuscript should be improved by making the following corrections and modifications before accepted for publishing.

Comments.

1.      How authors assign the targets of control antibiotics as the target of the peptides in question (line 124). what was the rationale for this assignment of the target of the molecule? (It will be useful to describe rationale of the algorithm (in CAMP database))

2.      The peptides were identified from saline environments, so it would be interesting to test their activity in saline conditions and compare the results to the MIC values obtained in this study.

3.      Most AMPs work by disrupting the bacterial membrane, especially amphipathic helical peptides. It would be helpful to confirm that the peptides identified in this study also have membrane activity. This could be done by measuring the killing kinetics of the peptides in parallel with the control antibiotics. Alternatively, D-enantiomers could be used to test the stereospecific binding of the peptides.

4.      The authors used in silico prediction models to predict the secondary structure of the peptides. It would be helpful to discuss the specific secondary structures that were observed and their implications. Additionally, instead of showing figures of the target proteins with highlighted hydrogen bonding residues, the authors could tabulate this information to save space and make it easier to compare. The space saved could then be used to depict the interaction of the peptides with the target proteins (3D models). This would provide more information about the structural features of the peptide-target interaction.

5.      The control antibiotic's MIC values should be included in Table 1 for easier comparison.

6.      MIC50 values should be represented as MIC50. Also, please correct to a more appropriate notation of µg/mL than µg/ml. Finally, please correct 0,25 (line 163 for consistency in the notations.

7.      The readability of the manuscript could be improved by choosing more appropriate words in some places. For example, the word "minor" (line 89) could be replaced with "limited". Additionally, the word "single use" (line 156) could be replaced with "alone".

8.      Table 1 and 5 could be merged to avoid the redundancy of information.

Please see the comments attached. 

Author Response

(The authors gave the same response as above.)

Round 2

Reviewer 1 Report

1. Manuscript has been corrected as suggested. Thanks to authors for this effort.

2. Authors should better discuss why high concentrations of identified AMP are acceptable from their point of view. I still fill uncomfortable with this question even if I agree that optimization could improve the MIC50.

3. A quick BLAST with the 3 provided peptide sequences provided very few number of hits. In every case, provided peptides seems to be fragments of bigger ones. In any case, obtained hits, their location and activity may be included in the discussion.
3.1. since sequences were issued form metagenomic data assembly, what is the real risk that there are mis-assembled fragments?
3.2. What could be the stability of a protein fragment in real life assays of pre-clinical tests?

4. Genbank submission should be completed before paper acceptance.

Author Response

We thank the Reviewer for comments. The answers have been sent in the attachment.

Reviewer 2 Report

The authors have implemented the recommended corrections and clarifications. I, therefore, recommend the manuscript for publication in its current state.

Author Response

We thank the Reviewer for comments that allowed us to improve the manuscript.